# Electron correlation and relativistic effects in the excited states of radium monofluoride

M. Athanasakis-Kaklamanakis [1,2,3] ✉, S. G. Wilkins [4,5] ✉, L. V. Skripnikov [25], Á. Koszorús[1,2], A. A. Breier[6,7], O. Ahmad[2], M. Au [8,9], S. W. Bai[10], I. Belošević[11], J. Berbalk [2], R. Berger [12], C. Bernerd[8], M. L. Bissell[13], A. Borschevsky[14], A. Brinson[4], K. Chrysalidis [8], T. E. Cocolios [2], R. P. de Groote[2], A. Dorne[2], C. M. Fajardo-Zambrano[2], R. W. Field [15], K. T. Flanagan[13,16], S. Franchoo [17,18], R. F. Garcia Ruiz [4,5], K. Gaul[12], S. Geldhof [2], T. F. Giesen[7], D. Hanstorp [19], R. Heinke [8], P. Imgram[2], T. A. Isaev [25], A. A. Kyuberis[14], S. Kujanpää [20], L. Lalanne[1,2], P. Lassègues[2], J. Lim [3], Y. C. Liu[10], K. M. Lynch [13], A. McGlone[13], W. C. Mei[10], G. Neyens [2] ✉, M. Nichols[19], L. Nies [1], L. F. Pašteka [14,21], H. A. Perrett[13], A. Raggio [20], J. R. Reilly[13], S. Rothe [8], E. Smets[2], S.-M. Udrescu[4], B. van den Borne [2], Q. Wang[22], J. Warbinek [23,24], J. Wessolek [8,13], X. F. Yang [10] & C. Zülch[12]

Highly accurate and precise electronic structure calculations of heavy radioactive atoms and their molecules are important for several research areas, including chemical, nuclear, and particle physics. Ab initio quantum chemistry can elucidate structural details in these systems that emerge from the interplay of relativistic and electron correlation effects, but the large number of electrons complicates the calculations, and the scarcity of experiments prevents insightful theory-experiment comparisons. Here we report the spectroscopy of the 14 lowest excited electronic states in the radioactive molecule radium monofluoride (RaF), which is proposed as a sensitive probe for searches of new physics. The observed excitation energies are compared with state-of-the-art relativistic Fock-space coupled cluster calculations, which achieve an agreement of ≥99.64% (within ~12 meV) with experiment for all states. Guided by theory, a firm assignment of the angular momentum and term symbol is made for 10 states and a tentative assignment for 4 states. The role of high-order electron correlation and quantum electrodynamics effects in the excitation energies is studied and found to be important for all states.

Achieving high-performance relativistic ab initio calculations of heavy, many-electron molecules with very high precision and accuracy is a critical milestone for several research areas at the intersection of particle, nuclear, atomic, and molecular physics. This is particularly important for elements that are difficult to study experimentally or for properties that are non-observable in the laboratory. Such research areas include, among others, superheavy element research[1], actinide chemistry[2,3], and searches for new physics with heavy molecules[4,5], which primarily rely upon the production of radioactive elements at dedicated accelerator facilities.

To assess the performance of ab initio computational chemistry, benchmarks using experimental measurements across a wide range of observables are of high importance. Laser spectroscopy offers a powerful avenue to study the electronic structure of radioactive

atoms[6], and molecules containing heavy atoms like those in the actinide series[7–9]. However, for molecules that contain atoms of the seventh period of the periodic table−with the exception of the quasistable [238]U and [232]Th−the availability of experimental measurements is significantly hindered by the radioactivity of the heavy nucleus.

As the last pre-actinide element, the electronic structures of radium compounds offer a powerful testing ground for the performance of ab initio quantum chemistry. Radium monofluoride (RaF) in particular has also received a lot of attention due to its promise as a sensitive probe for searches of new physics[10,11]. Such experiments aim to understand the limitations of the Standard Model (SM) and to assess the validity of candidate theories beyond the SM. To this end, among other approaches, precision tests of the SM and searches for new physics using atomic and molecular spectroscopy are being pursued[4], for instance to search for the symmetry-violating nuclear Schiff moment[12,13] or the electric dipole moment of the electron (eEDM)[14–16].

Due to the high degree of precision required by such experiments[5], ongoing and future campaigns are focused on systems with maximum sensitivity to the presence of symmetry-violating moments. RaF is a particularly promising system as it is amenable to direct laser cooling[10,17], which can lead to a further increase in precision by orders of magnitude[18]. Moreover, the ground state of RaF is highly sensitive to nuclear spin-dependent parity- or time-reversal violation[10,19–21], depending on the chosen isotope of the octupole-deformed radium nucleus[22], as well as the eEDM[23–25].

Extracting values of the symmetry-violating moments from experimental searches requires the calculation of molecular constants that quantify the sensitivity of the molecule to the moment of interest. Both in atoms and molecules[4,26], the theoretical precision and accuracy of the calculated molecular parameters will dictate the limit to which the symmetry-violating moment can be determined. As these sensitivity parameters are not experimentally measurable, benchmarking and improving the accuracy and precision of ab initio molecular theory across other observables, which can be measured in the laboratory, is also a necessary step towards precision tests of the SM[26,27]. Therefore, joint experimental and theoretical efforts have been devoted to evaluating the performance of state-of-the-art ab initio methods for many different properties of the structure of RaF.

All isotopes of radium have half-lives from nanoseconds to at most a few days, except for [226]Ra and [228]Ra (1600 and 5.75 years, respectively). These two long-lived isotopes have zero nuclear spin and are therefore not suited for the study of symmetry-violating nuclear moments. Radioactive ion beam (RIB) facilities are favorable not only for the preparatory spectroscopic studies needed to understand the electronic structure of the different isotopologues of RaF, but also for future precision experiments. The first spectroscopic studies on RaF molecules were performed at the CERN-ISOLDE radioactive beam facility. This resulted in initial insight into the low-energy electronic-vibrational structure of RaF[11], the observation of a strong isotope shift across several short-lived isotopologues[28], and a realistic laser-cooling scheme[17].

The initial experiment and the interpretation of the data were driven by prior quantum chemistry calculations of the electronic structure of RaF[11]. Subsequent theoretical studies with single-reference coupled cluster theory including a higher-level treatment of electron correlations and quantum electrodynamic (QED) effects[29] suggest a re-evaluation of some of the previous spectroscopic assignments[11]. Furthermore, the very high precision that the calculations achieved for the prediction of low-lying excited-state energies, with an uncertainty of only a few tens of cm$^{-1}$ (few meV)[29,30], call for experimental verification of their accuracy.

This work reports the observation of all 14 excited electronic states in RaF that are predicted to exist up to 30,000 cm$^{-1}$ above the ground state. The observed excitation energies are compared to relativistic state-of-the-art Fock-space coupled cluster (FS-RCC)

calculations[31], including QED corrections and fully treated triple-cluster amplitudes that capture high-order electron correlation effects. The results highlight the power of FS-RCC for highly precise and accurate calculations of excitation energies in heavy molecules, even at high excitation energies. As a multi-reference approach, the FS-RCC method is also applicable to systems whose states have a multi-reference character[32,33], where single-reference coupled cluster theory is not applicable.

## Results

Figure 1 shows typical experimental spectra obtained in this study, and the rotational branch assignment determined from contour fitting. In Fig. 2, the experimentally observed excitation energies are compared with the predictions from several FS-RCC calculations at different levels of sophistication.

The discovery of the excited states of [226]RaF was guided by theoretical predictions from FS-RCC calculations with single- and double-excitation amplitudes (FS-RCCSD), using doubly augmented (aug) Dyall CV4Z basis sets[34,35] and correlating 27 electrons (27e-augCV4Z) within the Dirac-Coulomb Hamiltonian. Such calculations can be completed within a few days and are therefore well-suited to guide the experimental efforts. Although such calculations have a limited accuracy for high-lying states (within hundreds of cm$^{-1}$) it is sufficient to direct the experimental search to the correct energy range, leading to the discovery of 10 new excited states (teal lines in Fig. 2 as well as $B^2\Delta_{5/2}$) in this work.

To study the accuracy of the electronic structure ab-initio methods as a function of excitation energy, further FS-RCCSD calculations were performed at an extended level of correlation treatment, using enhanced basis sets, and an improved electronic Hamiltonian compared to the 27e-augCV4Z calculations that guided the experiment. The agreement between the observed level energies and the most

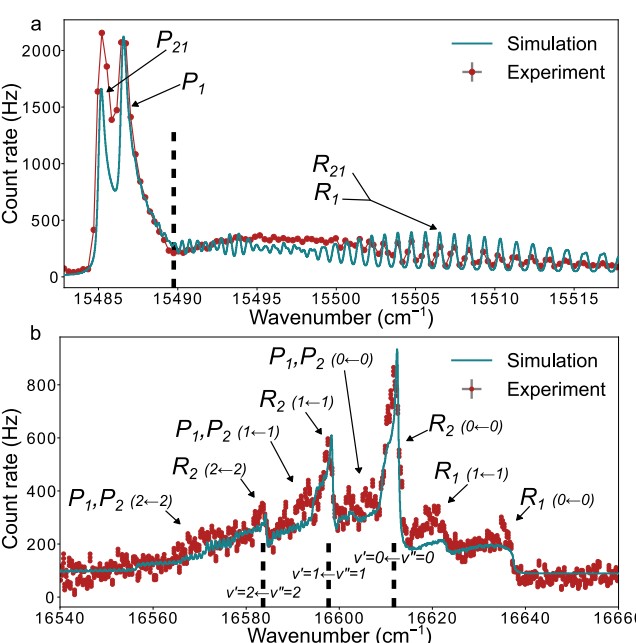

**Fig. 1 | Example spectra.** **a** $G^2\Pi_{1/2} \leftarrow A^2\Pi_{1/2}$ ($v' = 0 \leftarrow v'' = 0$). **b** $C^2\Sigma_{1/2} \leftarrow X^2\Sigma_{1/2}$, showing multiple $\Delta v = 0$ bands. The simulated spectra were constructed using the best-fit molecular parameters determined from contour fitting with PGOPHER[78]. The $x$-axis corresponds to the wavenumber of the scanning laser. The rotational branches are noted on the plots, and the spectral features marked by a dashed line denote the band heads. Where branch labels lead to the same arrow, the branches cannot be resolved. The parentheses denote the vibrational transition that the rotational branch belongs to. Error bars correspond to the statistical uncertainty of the count rate at each point.

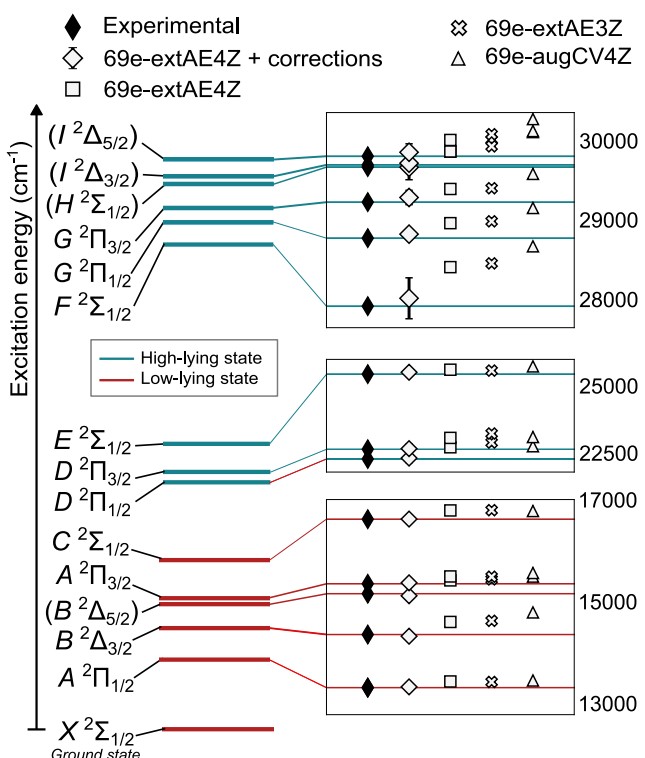

**Fig. 2 | Level diagram of RaF up to 30,000 cm⁻¹.** Left: All levels are observed experimentally in this work. The electronic term symbols have been assigned according to the 69e-extAE4Z + corrections calculations (see text for details). Right: Zoomed-in comparison of experimental excitation energies with respect to the vibronic ground state and FS-RCCSD calculations at different levels of sophistication (increasing from right to left). The wavenumber scale pertains to each inset plot separately. Error bars are included only for the most precise calculations (wide diamonds) and correspond to the uncertainty of the method of calculations, as described in *Methods*. In most cases, the error bars are smaller than the data marker.

**Table 1 | Comparison of experimental and theoretical electronic excitation energies (*T*₀, in cm⁻¹) in RaF**

| State | Experiment | Theory | Agreement |
|---|---|---|---|
| $X^2\Sigma_{1/2}$ | 0 | 0 | |
| $A^2\Pi_{1/2}$ | 13,284.427(1)$_{stat}$(20)$_{syst}$ ᵃ | 13,299(36) | 99.89 |
| $B^2\Delta_{3/2}$ | 14,333.00(161)$_{stat}$(51)$_{syst}$ | 14,300(61) | 99.77 |
| $(B^2\Delta_{5/2})$ | 15,140.36(48)$_{stat}$(51)$_{syst}$ | 15,099(70) | 99.73 |
| $A^2\Pi_{3/2}$ | 15,335.73(49)$_{stat}$(62)$_{syst}$ | 15,355(35) | 99.87 |
| $C^2\Sigma_{1/2}$ | 16,612.06(18)$_{stat}$(51)$_{syst}$ | 16,615(69) | 99.98 |
| $D^2\Pi_{1/2}$ | 22,289.47(29)$_{stat}$(51)$_{syst}$ | 22,320(169) | 99.86 |
| $D^2\Pi_{3/2}$ | 22,651.09(31)$_{stat}$(51)$_{syst}$ | 22,673(170) | 99.90 |
| $E^2\Sigma_{1/2}$ | 25,451.12(11)$_{stat}$(26)$_{syst}$ | 25,520(84) | 99.73 |
| $F^2\Sigma_{1/2}$ | 27,919.57(180)$_{stat}$(51)$_{syst}$ | 28,019(257) | 99.64 |
| $G^2\Pi_{1/2}$ | 28,774.07(51)$_{stat}$(35)$_{syst}$ | 28,824(111) | 99.83 |
| $G^2\Pi_{3/2}$ | 29,225.64(25)$_{stat}$(51)$_{syst}$ | 29,284(90) | 99.80 |
| $(H^2\Sigma_{1/2})$ | 29,665.54(67)$_{stat}$(51)$_{syst}$ | 29,663(156) | 99.99 |
| $(I^2\Delta_{3/2})$ | 29,693.15(24)$_{stat}$(51)$_{syst}$ | 29,715(102) | 99.92 |
| $(I^2\Delta_{5/2})$ | 29,801.59(7)$_{stat}$(35)$_{syst}$ | 29,852(106) | 99.83 |

The theoretical values correspond to the 69e-extAE4Z calculations with 27e-T, CBS, Gaunt, and QED corrections (wide diamonds in Fig. 2). The normalized theoretical agreement (%) is reported as $1 - \frac{|E_{th} - E_{exp}|}{E_{exp}}$. Parentheses denote tentative assignment. Statistical and systematic errors are noted next to the corresponding brackets and defined in the Supplementary Information.
ᵃValue from ref. [17].

advanced calculations allowed a simplification of the electronic-state assignments.

To improve the treatment of electron correlations, the correlation space was expanded to include 69 electrons. Adding the remaining 28 electrons that correspond to the 1s–3d shells of Ra, thus including all 97 RaF electrons in the correlation space, modified the level energies by up to 2 cm⁻¹ (see Supplementary Table III), which is significantly below the total theoretical uncertainty, reported in Table 1 for each state.

To improve the basis-set quality, calculations were performed with the extended (ext) AE3Z[36,37] (crosses in Fig. 2) and AE4Z[30,37] (squares in Fig. 2) basis sets, which include a greater number of functions for a more accurate description of the electronic states. A further correction for the incompleteness of the basis sets (CBS correction) was implemented based on the scalar-relativistic treatment of valence and outer-core electrons[38–40] (see *Methods* for more details).

The accuracy of the electronic Hamiltonian was improved by taking into account the Gaunt inter-electron interaction[41] and QED effects[42], with the latter made possible recently for molecular 4-component calculations[30]. Lastly, higher-order electron correlation effects encoded in the triple-excitation amplitudes (T) were included via the FS-RCCSDT approach[31]. The challenging task of simultaneously calculating the triple-excitation contribution to the excitation energies for 15 molecular states that have different electronic configurations was feasible thanks to the use of compact relativistic basis sets[36,43,44], developed for use with the 2-component generalized relativistic effective-core potential (GRECP) as the Hamiltonian[38–40]. The triple-

excitation amplitudes were calculated for the 27 outermost electrons (correction denoted as 27e-T), including down to the 5d radium electrons.

The final theoretical values taking into account CBS, Gaunt, QED, and 27e-T corrections are included in the wide diamond markers displayed in Fig. 2 and compared to the experimental excitation energies in Table 1. The contribution of different corrections to the final theoretical transition energies are provided in Supplementary Table II.

An overall agreement of at least 99.64% is achieved for all states, which allowed assigning the newly observed states and revising earlier tentative assignments. A transition observed at 16,175.2(5) cm⁻¹ in ref. 11 was previously assigned as $C^2\Sigma_{1/2} \leftarrow X^2\Sigma_{1/2}(v' = 0 \leftarrow v'' = 0)$. The theoretical precision achieved in the present study together with the new measurements indicate that this transition does not correspond to the lowest electronic excitation energy of the upper state, but rather corresponds to the $(v' = 0 \leftarrow v'' = 1)$ vibrational transition, as suggested in ref. 29. Instead, a new spectrum observed in this work very close to the predicted value of 16,615 cm⁻¹ is identified as the $v' = 0 \leftarrow v'' = 0$ vibronic transition to the $C^2\Sigma_{1/2}$ state. Additionally, a transition observed at 15,142.7(5) cm⁻¹ in ref. 11 was previously tentatively assigned as $(B^2\Delta_{3/2}) \leftarrow X^2\Sigma_{1/2}(v' = 0 \leftarrow v'' = 0)$. The close agreement of this transition energy, observed in this work to lie at 15,140.36(48)$_{stat}$(51)$_{syst}$ cm⁻¹, with the calculated value of 15,099 cm⁻¹ leads to the assignment of this transition as $(B^2\Delta_{5/2}) \leftarrow X^2\Sigma_{1/2}$ $(v' = 0 \leftarrow v'' = 0)$. Finally, a new transition observed at 14,333.00(161)$_{stat}$(51)$_{syst}$ cm⁻¹ is closer to the theoretical prediction for the excitation energy of the $B^2\Delta_{3/2}$ state, and is in agreement with the predictions in ref. 29. Thus, the assignment of the $B^2\Delta_{3/2}$ state at 14,333.00(161)$_{stat}$(51)$_{syst}$ cm⁻¹ is adopted. The term assignments for the newly observed high-lying states (teal lines in Fig. 2), lying above 20,000 cm⁻¹ from the ground state, are possible with the highly accurate ab initio calculations.

Figure 3a, b present a detailed comparison of the impact of each of the corrections discussed above on the experiment-theory agreement for all states. In particular, the impact of treating triple-excitation amplitudes at high electronic excitation energies is clearly visible in Fig. 3b. The 27e-T correction has the most prominent effect in

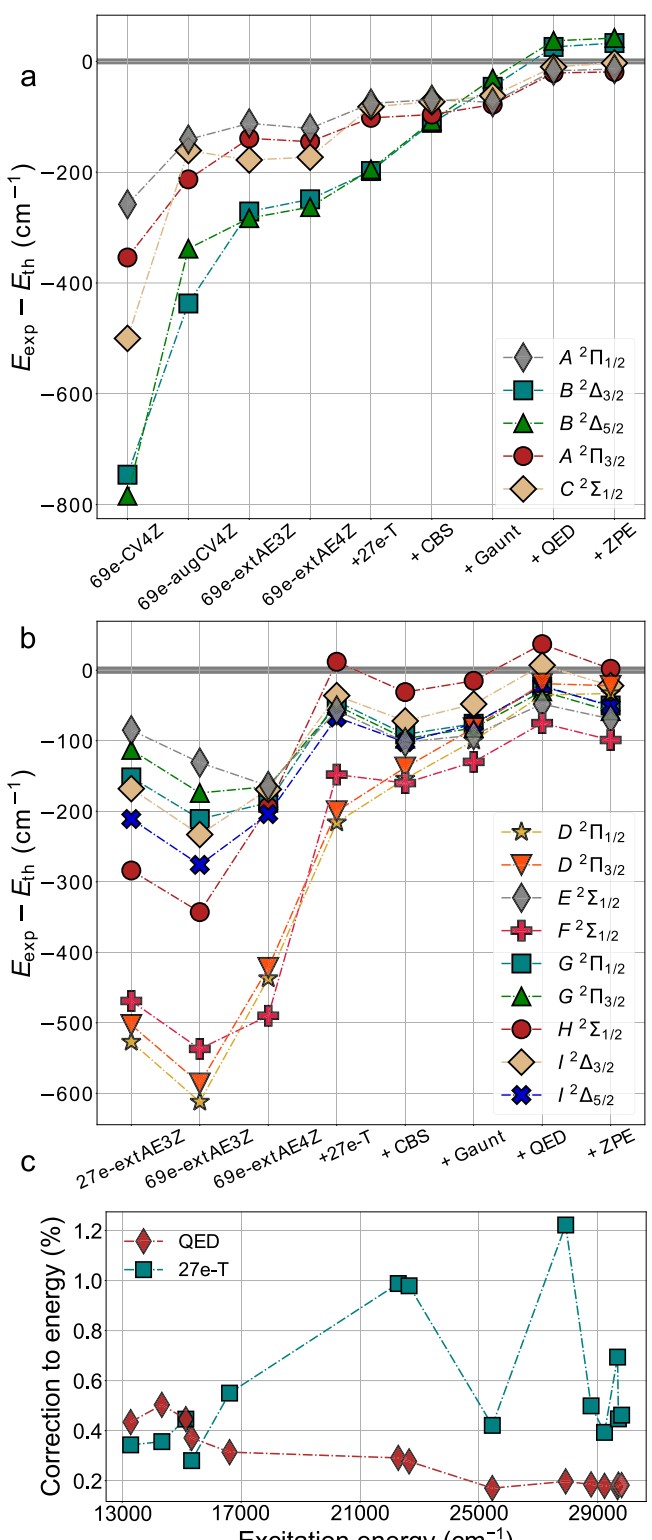

**Fig. 3 | Comparison of theory and experiment.** Evolution of experiment-theory agreement as a function of increasing theoretical sophistication for (**a**) the five lowest-lying states and (**b**) the nine high-lying states. For the high-lying states, the CV4Z results did not allow relating calculated and observed levels, and they are not included. '+ZPE' corresponds to the zero-point vibrational energy correction. **c** Evolution of the QED and 27e-T corrections to the excitation energies calculated at the 69e-extAE4Z level.

improving the agreement with experiment among all listed corrections and for all considered high-lying states. This correction is larger in high-lying states (Fig. 3b) than in the low-lying ones (Fig. 3a), demonstrating the need for spectroscopic studies of electronic states far above the ground state to understand the role of correlations (and relaxation) in the electronic structure. Figure 3a also demonstrates the importance of choosing an appropriate basis set for calculations of excited electronic states even energetically close to the ground state, as the difference between 69e-extAE4Z and 69e-CV4Z with the unmodified CV4Z basis sets[34,35] is considerable for all states.

In Fig. 3b, the increase of the correlation space from 27 to 69 electrons using the same basis set leads to an apparent decrease in the agreement with experiment for all states. This can be attributed to a case of mutually canceling errors when a smaller correlation space is used.

Finally, in Fig. 3c the impact of including QED contributions and an iterative treatment of triple-excitation amplitudes is presented. The contribution of QED effects was found to be especially important for the low-lying states—excitation energies up to 20,000 cm⁻¹—having a greater effect on improving the agreement with experiment than the iterative treatment of triple-excitation amplitudes in the FS-RCC models.

Figure 3c confirms that the relative impact of QED effects is indeed significant at low energies, but it decreases in importance for greater excitation energies. On the other hand, the higher-order electron correlation effects captured by the iterative treatment of triple-excitation amplitudes are of increasing importance for higher-energy states, but remain non-negligible energetically close to the ground state. This is explained by the participation of non-valence outer-core electrons in the higher-energy excitations. Specifically for the excitation energies of the high-lying states, it is found that the contribution of $5d$ electrons plays an important role (see Supplementary Table III). The 27e-T correction is most pronounced particularly for the $D$ and $F$ states, where it reaches the 1% level.

As in the lighter alkaline-earth monohalides, all bound electronic states in RaF belong to Rydberg series of an unpaired electron centered on a RaF⁺ core, which converge to the ground state of RaF⁺. The calculated composition of the orbitals occupied by the unpaired electron is given in Table 2. This simple picture provides a way to understand the electronic structure of the molecule even at high excitation energy, with non-relativistic multichannel quantum defect theory based on this picture being very successful for the lighter homologs[45,46].

Using the measured ionization potential (IP) of RaF at $4.969(2)_{stat}(10)_{syst}$ eV[47], the effective principal quantum number $n^*$ can be extracted for every state in RaF as:

$$n^* = \sqrt{\frac{R}{IP - E}} \qquad (1)$$

where $R$ is the Rydberg constant, $E$ is the excitation energy of the state, and $n^* = n - \mu$ with $n$ being the integer principal quantum number and $\mu$ the quantum defect. The $n^*$ values are given in Table 3, and a plot of $n^*$ mod(1) versus $n^*$ is shown in Fig. 4a. Similar to BaF, a total of 10 core-penetrating Rydberg series are expected in RaF—four ²Σ, three ²Π, two ²Δ, and one ²Φ series—arising from the mixture of the $s$, $p$, $d$, and $f$ Rydberg series in Ra⁺ into a core-penetrating, strongly $l$-mixed $s$ - $p$ - $d$ - $f$ supercomplex[48]. In Fig. 4a, lines connect states that belong to the same Rydberg series, based on having the same $\Lambda$ and similar $n^*$ mod(1)[49]. The start of four Rydberg series are identified, two ²Σ, one ²Π, and one ²Δ, with two more series starting with $D$²Π and $F$²Σ. Observing all Rydberg series in the $s$ - $p$ - $d$ - $f$ supercomplex requires further spectroscopy at higher excitation energy in the future.

**Table 2 | Composition of states in RaF in terms of Ra⁺ valence electron configurations**

| State | Composition |
|---|---|
| $X^2\Sigma_{1/2}$ | 90%7s |
| $A^2\Pi_{1/2}$ | 60%7$p_{1/2}$ + 20%6$d_{3/2}$ + 10%7$p_{3/2}$ |
| $B^2\Delta_{3/2}$ | 40%6$d_{3/2}$ + 30%6$d_{5/2}$ + 20%7$p_{3/2}$ |
| $B^2\Delta_{5/2}$ | 90%6$d_{5/2}$ + 10%7$d_{5/2}$ |
| $A^2\Pi_{3/2}$ | 50%7$p_{3/2}$ + 40%6$d_{3/2}$ |
| $C^2\Sigma_{1/2}$ | 50%7$p_{3/2}$ + 30%6$d_{5/2}$ + 10%7$d_{5/2}$ |
| $D^2\Pi_{1/2}$ | 40%6$d_{3/2}$ + 20%7$p_{1/2}$ + 20%6$d_{5/2}$ + 10%7$p_{1/2}$ + 10%8$p_{1/2}$ |
| $D^2\Pi_{3/2}$ | 50%6$d_{5/2}$ + 30%7$p_{3/2}$ + 10%8$p_{3/2}$ + 10%6$d_{3/2}$ + 10%7$d_{5/2}$ |
| $E^2\Sigma_{1/2}$ | 70%8s + 10%8$p_{1/2}$ + 10%9s |
| $F^2\Sigma_{1/2}$ | 30%8$p_{1/2}$ + 20%8$p_{3/2}$ + 10%7$p_{3/2}$ + 10%6$d_{3/2}$ + 10%6$d_{5/2}$ + 10%8s |
| $G^2\Pi_{1/2}$ | 30%8$p_{3/2}$ + 30%7$d_{3/2}$ + 20%8$p_{1/2}$ + 10%7$p_{1/2}$ |
| $G^2\Pi_{3/2}$ | 50%8$p_{3/2}$ + 20%7$d_{5/2}$ + 10%7$p_{3/2}$ + 10 %7$d_{3/2}$ |
| $H^2\Sigma_{1/2}$ | 40%7$d_{5/2}$ + 10%7$p_{3/2}$ + 10%8$p_{3/2}$ + 10%7$d_{3/2}$ |
| $I^2\Delta_{3/2}$ | 60%7$p_{3/2}$ + 20%7$d_{5/2}$ |
| $I^2\Delta_{5/2}$ | 70%7$d_{5/2}$ + 10%6$d_{5/2}$ |

The composition is calculated as the mean value of the projectors onto the one-electron atomic orbitals of the Ra⁺ cation over the FS-RCCSD wave function of RaF (see *Methods*). Only contributions with a relative impact of ≥10% are shown, and contributions are rounded.

**Table 3 | Observed spin-orbit interaction constants *A* and effective principal quantum numbers *n*\* for the states in RaF assigned in this work**

| | *A* (cm⁻¹) | *n*\* |
|---|---|---|
| $X^2\Sigma$ | | 1.65 |
| $A^2\Pi$ | 2051(1) | 2.06 |
| $B^2\Delta$ | [404(1)] | 2.08 |
| $C^2\Sigma$ | | 2.16 |
| $D^2\Pi$ | 362(1) | 2.50 |
| $E^2\Sigma$ | | 2.74 |
| $F^2\Sigma$ | | 3.00 |
| $G^2\Pi$ | 452(1) | 3.15 |
| $H^2\Sigma$ | | 3.25 |
| $I^2\Delta$ | [54(1)] | 3.26 |

The constants for $B^2\Delta$ and $I^2\Delta$ are tentative and shown in brackets.

The bond in alkaline-earth monohalides is well-understood[50,51] to be the result of the electrostatic interaction of a metal cation and a halogen anion, with the valence molecular electron being strongly localized on the metal cation. In RaF, the electric field of F⁻ polarizes the $^2L$ states of Ra⁺, causing $\Delta L = \pm 1$, $\Delta\lambda = 0$ mixing, resulting in $L$, $\lambda$-dependent energy shifts and Ra⁺-to-RaF modification of spin-orbit (SO) constants. These effects are dependent on Ra⁺ electric dipole transition moments, the internuclear distance, and $\Delta L = \pm 1$ energy differences of the free Ra⁺ ion. These polarization effects are responsible for level shifts, level splittings, and configuration mixing, which could be captured by a de-perturbation model for all radium monohalides[50].

The observed SO constants are shown in Table 3. The SO constants for the *A* and *G* states as well as the *B* and *I* states, which are part of the same $^2\Pi$ and $^2\Delta$ Rydberg series, exhibit reasonable agreement with the expected $(n^*)^{-3}$ scaling[49]. The SO constants in RaF differ from those in Ra⁺ due to the field-induced mixing. This effect increases with

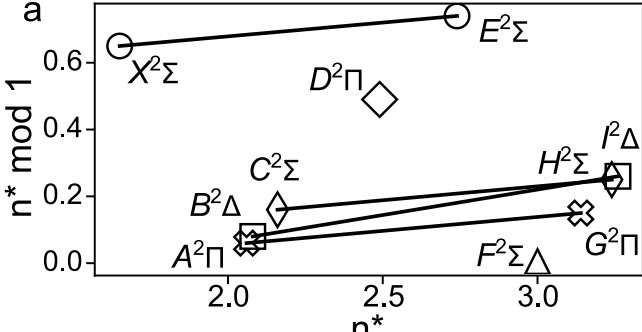

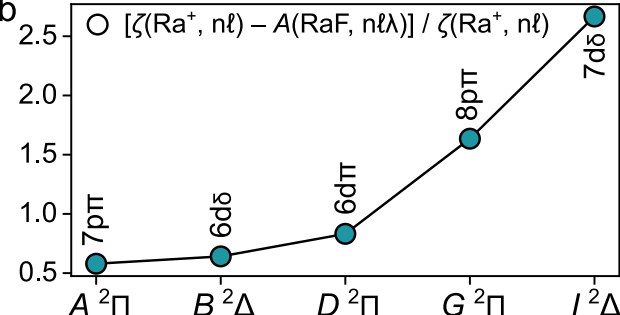

**Fig. 4 | Rydberg series in RaF. a** Effective principal quantum numbers *n*\* and their residuals modulo 1 for excited states in RaF. Lines connect states that belong to the same Rydberg series, based on having the same $\Lambda$ and similar *n*\* mod. **b** Normalized difference in SO constant between states in RaF and Ra⁺ with corresponding (leading) configurations. The symbol $\zeta$ corresponds to the microscopic SO constant for atomic states with $n\ell$ valence-electron configuration, and *A* to the molecular SO constant with $n\ell\lambda$ valence-electron configuration.

excitation energy, as seen in Fig. 4b, because of the increase in the density of Ra⁺ electronic states. The ab initio calculations reflect this energy-dependent configuration mixing, shown as an increase in the number of configurations that contribute >10% to the composition of electronic states above $C^2\Sigma_{1/2}$, see Table 2.

In terms of the overall computational precision in this work, it is noted that the absolute difference between theory and experiment in Table 1 is significantly smaller than the theoretical uncertainty for all states, which was determined with an established uncertainty-estimation scheme. The experimental benchmark in Table 1 highlights that the scheme is conservative and the theoretical uncertainty derived from it corresponds more closely to a $2\sigma$ error. If the agreement between experiment and theory in Table 1 is interpreted as a measure of the theoretical precision—that is, a quantification of the true theoretical uncertainty—then the true $1\sigma$ error of the calculations amounts to ≤0.2% of the excitation energy for all states. This highlights the high precision that FS-RCC calculations can achieve.

A highly accurate and precise treatment of electron correlation in RaF and its reliable uncertainty estimation may also be important for the efforts to calculate the sensitivity of molecular electronic states to nuclear, hadronic, and leptonic symmetry-violating moments, where no experimental verification of computed molecular parameters is possible. All previous theoretical studies of the sensitivity to different symmetry-violating moments in RaF[23–25,52–54] have reported results either using CCSD theory (with triple-excitation amplitudes included only via approximations in some works), or using the Zeroth-Order Regular Approximation based on a mean-field approach and density functional theory, which do not fully capture correlation and relativistic effects.

Figure 2 and Table 1 show that the 69e-extAE4Z calculations with 27e-T, CBS, Gaunt, and QED corrections reproduce all experimentally observed energies with a deviation better than 0.5%, which surpasses

that of all previous relativistic FS-RCCSD calculations of alkaline-earth monofluorides (refs. 55–57), while the calculated excitation energies have an uncertainty of <1%. Therefore, simultaneously high accuracy and precision is reached even for states energetically far from the ground state.

The achieved agreement justifies the assigned angular momenta and term symbols for the observed levels, leading to a significantly expanded electronic map of RaF. The number of firmly assigned states in this work advances the understanding of the electronic structure of RaF to be on par with that of uranium and thorium molecules[8,9,58,59], whose experimental study is not constrained by scarcity or radio-activity. Extensions of the present work to search for states above 30,000 cm$^{-1}$ from the ground state can allow the further assignment of the electronic states in RaF into Rydberg series, which will elucidate the evolution of quantum defects and core-valence electron interactions across the alkaline-earth monofluorides[48,60].

The present study, both experimental and theoretical, paves the way for future high-resolution studies of these states and tests the predictive power of the calculations, whose reliability is a prerequisite for future precision tests of the SM and other areas of fundamental and applied science. Moreover, the performance of FS-RCC in this work highlights the method's suitability for predictions of electronic transitions in wide range of species, such as superheavy atoms and polyatomic molecules, where accurate and precise predictions of the excitation energies are critical for the successful discovery of electronic transitions

## Methods
### Experiment
Laser spectroscopy of $^{226}$RaF was performed using the Collinear Resonance Ionization Spectroscopy (CRIS) experiment at CERN-ISOLDE.

Accelerated beams of $^{226}$RaF$^+$ were produced at the CERN-ISOLDE RIB facility[61]. Two weeks prior to the experiment, short- and long-lived radioactive isotopes, among which $^{226}$Ra nuclei ($t_{1/2} = 1600$ years), were produced by impinging 1.4-GeV protons onto a room-temperature uranium carbide target. During the experiment, the irradiated target was gradually heated up to 2000 °C to extract the produced radio-nuclides from within the solid matrix. Due to the asynchronous radionuclide production and extraction from the target, which was chosen so as to suppress beam contaminants with short half-lives, the extraction rate for a given temperature was gradually decreasing over time. At regular intervals, the target temperature was increased in a controlled manner to recover a consistent extraction rate.

By exposing the target to a constant flow of CF$_4$, the radium atoms formed $^{226}$Ra$^{19}$F molecules that were ionized using a rhenium surface ion source. The $^{226}$RaF$^+$ ions were then accelerated to 40 keV and mass-separated from other radiogenic species using two magnetic dipole separators. The continuous, isotopically pure beam of $^{226}$RaF$^+$ was then accumulated in a radiofrequency quadrupolar cooler-buncher (RFQcb), which released the $^{226}$RaF$^+$ beam at a kinetic energy of 39,900 eV in bunches with a 5-$\mu$s temporal spread once every 20 ms. The ion beam was identified as composed purely of $^{226}$Ra$^{19}$F$^+$ using the ISOLTRAP multi-reflection time-of-flight mass spectrometer[62].

The internal temperature of the beam was cooled to near room temperature while being trapped in the RFQcb in the presence of a helium buffer gas. The temperature at which the ensemble delivered from the RFQcb thermalized was found to vary over time, ranging from 350 to 600 K. In all cases, the spectra could be fitted assuming that the molecular ensemble had a uniform temperature distribution. Therefore, it is considered that above a certain target temperature, the produced molecules could not be cooled to room temperature within the time they were trapped in the RFQcb, but thermalized at a temperature between the target environment and the room-temperature buffer gas.

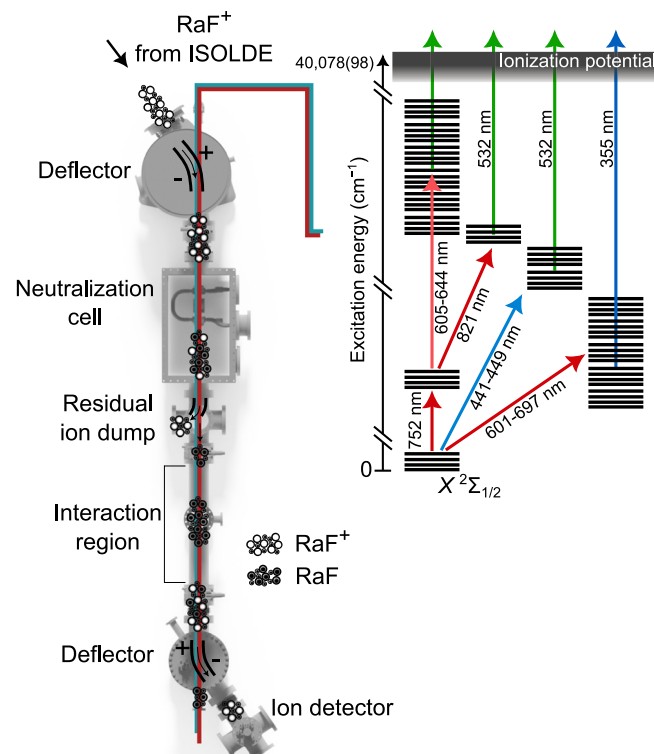

**Fig. 5 | Schematic of the CRIS technique and the laser schemes used for resonance ionization spectroscopy of RaF in this work.**

A typical $^{226}$RaF$^+$ beam intensity of $1.2 \times 10^6$ ions per second was then sent into the CRIS beam line, which is shown schematically in Fig. 5. At CRIS, the fast ion beam was firstly neutralized via collisions with hot sodium vapor in a charge-exchange cell, at the exit of which all the ions that were not successfully neutralized were deflected onto a beam dump. The bunches of neutralized molecules entered the interaction region, where they interacted collinearly with pulsed laser beams that step-wise excited the molecular electrons from the ground state to above the IP. The final laser step in the CRIS scheme induced non-resonant ionization of the molecules that were resonantly brought to an excited state by the preceding laser steps. As a result, successful ionization of the neutralized molecules required one (in two-step schemes) or two (in three-step schemes) resonant laser excitations. At the end of the interaction region, the resonantly ionized molecules were deflected away from the residual neutral particles and onto a single ion counter using electrostatic deflectors. The molecular spectra were obtained by monitoring the ion count rate as a function of the laser frequencies of the resonant excitation steps.

### Laser setup
All lasers used in this work were pulsed and overlapped with the molecular beam in a collinear geometry. Multiple laser schemes were used for the spectroscopy of the excited states in RaF. For each scheme, the molecules underwent either one or two resonant excitations starting from the electronic ground state using tunable pulsed titanium-sapphire (Ti:Sa) or pulsed dye lasers (PDLs, Spectra Physics PDL and Sirah Cobra). A high-power 532-nm or 355-nm Nd:YAG laser was used to ionize the molecules that had been resonantly excited. The molecular excitation energies were measured by scanning the frequency of the tunable laser used for a resonant transition while monitoring the ion count rate.

The level search was facilitated by preparing multiple laser-ionization schemes based on two PDLs with nominal linewidths $\Delta f$ of 0.8 and 9 GHz and two grating-based broadband pulsed Ti:Sa lasers

($\Delta f \approx 3$ GHz). A total wavenumber range $>4000$ cm$^{-1}$ was scanned in a period of a few days. Further information about the data analysis and spectroscopic assignment can be found in the Supplementary Information. More details on the experimental technique can be found in refs. 11,63.

The spectrum of the previously reported transition from the ground state to the $A^2\Pi_{1/2}$ state was measured using a grating-based titanium:sapphire (Ti:Sa) laser scanned around 752 nm (in the molecular rest frame), corresponding to the transition from the ground state. The transition from the electronic ground state to this level was then used as the first step in three-step schemes to search for most high-lying levels, using either its $v' = 0 \leftarrow v'' = 0$ or $v' = 1 \leftarrow v'' = 1$ vibrational transitions.

The spectra of the transitions to the $B^2\Delta_{3/2}$, $B^2\Delta_{5/2}$, $A^2\Pi_{3/2}$, and $C^2\Sigma_{1/2}$ states were measured with two-step schemes by scanning a dye laser around 697 nm (Pyridine 1) for $B^2\Delta_{3/2}$, 661 nm (DCM) for $B^2\Delta_{5/2}$, 651 nm (DCM) for $A^2\Pi_{3/2}$, and 601 nm (Pyridine 1) for $C^2\Sigma_{1/2}$, followed by a 355-nm non-resonant ionization step. The $D^2\Pi$ states were also measured with a two-step scheme using second-harmonic Ti:Sa for the resonant excitation from the $X^2\Sigma_{1/2}$ ground state, followed by non-resonant ionization using a 532-nm Nd:YAG laser.

The remaining high-lying states were observed using three-step schemes, where the first step was the resonant transition from $X^2\Sigma_{1/2}$ to $A^2\Pi_{1/2}$ ($0 \leftarrow 0$ or $1 \leftarrow 1$), followed by the resonant excitation to the high-lying states. The ionization was induced by a non-resonant excitation driven by a 532-nm Nd:YAG laser. To search for the transition to the $E^2\Sigma_{1/2}$ state, the second laser step was scanned around 821 nm with a second grating-based Ti:Sa laser. For the transitions to the $G^2\Pi_{1/2}$, $G^2\Pi_{3/2}$, $H^2\Sigma_{1/2}$, $I^2\Delta_{3/2}$, and $I^2\Delta_{5/2}$ states, the second laser-excitation step was scanned around 644 nm, 625 nm, 610 nm, 609 nm, and 605 nm, respectively, using a dye laser (DCM). For the transition to the $F^2\Sigma_{1/2}$ state, a dye laser at 683 nm (Pyridine 1) was used. Figure 5 pictorially summarizes the laser schemes used in this work.

The Ti:Sa lasers were pumped by a 532-nm Nd:YAG laser operating at 1 kHz, while the dye lasers were pumped by 532-nm Nd:YAG lasers operating at 50 Hz. The 532-nm Nd:YAG laser used for non-resonant ionization was operating at 50 Hz, as well, while the 355-nm Nd:YAG laser was operating at 100 Hz. The relative timing between the laser steps was controlled by triggering the Q-switches of the pulsed lasers using a multi-channel, ultra-low-jitter clock (Quantum Composer 9528). As the excited-state lifetimes were not known for the newly discovered electronic states, all three steps were temporally overlapped. The frequencies of the dye lasers were measured using a HighFinesse WS6-600 wavemeter and the frequencies of the Ti:Sa lasers were measured using a HighFinesse WSU-2 wavemeter. The WSU-2 wavemeter was continuously calibrated by monitoring at the same time the frequency of a diode laser (Toptica DLpro) locked to a hyperfine peak in rubidium.

## Calculations

FS-RCCSD calculations were performed using the DIRAC[64,65] code. FS-RCCSDT calculations were performed using the EXP-T code[66,67]. Single reference relativistic coupled cluster calculations were performed with the MRCC[68,69] code. All scalar relativistic correlation calculations were performed using the public version of the CFOUR[70] code. Matrix elements of the model QED Hamiltonian were calculated within the code developed in ref. 30.

The electronic ground state of the RaF$^+$ cation was chosen as a Fermi-vacuum in the Fock-space (FS) calculation. Target states in the neutral RaF are considered as belonging to the one-particle sector of the FS. In the calculation, the Dirac-Coulomb Hamiltonian was used to solve the self-consistent (Dirac-Hartree-Fock) problem and then converted to the two-component all-electron Hamiltonian by means of the exact-two-component (X2C) technique within the molecular mean-field approximation[41]. For brevity, we refer to this as the X2C-DC

approach. For the 69-electron FS-RCCSD calculations, the extAE4Z basis set was used, which corresponds to the extended uncontracted Dyall's all-electron AE4Z basis set for Ra[35] from ref. 30 and the uncontracted Dyall's AAEQZ basis set[37] for F. Explicitly, this basis set includes $[42s\,38p\,27d\,27f\,13g\,3h\,2i]$ Gaussian-type functions for Ra. In these FS-RCCSD calculations, the energy cutoff for virtual orbitals was set to 300 Hartree to ensure the proper correlation of outer-core electrons. For the 69-electron FS-RCCSD calculations (employed in the initial discovery of the excited states) using a doubly augmented Dyall's CV4Z basis set[34,35], the energy cutoff for virtual orbitals was set to 100 Hartree.

In addition to the 69e-FS-RCCSD-extAE4Z calculations, further calculations with different numbers of correlated electrons and utilizing different basis sets were performed to probe various computational aspects. These included 27- and 69-electron calculations in the standard (unmodified) uncontracted Dyall's CV4Z basis set and 17-, 27-, 35-, 69-, and 97-electron calculations in the extended AE3Z basis set[34,35,37] (see Supplementary Tables III and IV). The latter extAE3Z basis set has been developed in ref. 36 and includes $[38s\,33p\,24d\,14f\,7g\,3h\,2i]$ Gaussian-type functions for Ra and corresponds to the uncontracted AE3Z[37] basis set on F.

To take into account contributions of an extended number of basis functions with high angular momentum ($l \geq 4$) beyond those contained in extAE4Z, the scalar-relativistic variant of the valence part of the generalized relativistic effective core potential approach[38–40] was used, as well as the 37e-EOMEA-CCSD approach (which is equivalent to FS-RCCSD in the considered case) to treat electron correlations using the CFOUR code[70]. In this way, it was possible to extend the basis set towards higher harmonics in Ra up to $15g\,15h\,15i$ (with an additional increase in the number of $s\,p\,d$ functions), which is intractable in practice within the Dirac-Coulomb Hamiltonian using available resources. Following ref. 30, the extrapolated contribution of higher harmonics to the basis-set correction was also included. In the extrapolation scheme, the contribution of basis functions with an angular momentum $l$ (for $l > 6$) is determined using the formula $A_1/l^5 + A_2/l^6$, where the coefficients $A_1$ and $A_2$ were derived from the directly calculated contributions of $h$- ($l = 5$) and $i$- ($l = 6$) harmonics. Thus, we calculated the sum of contributions for $l > 6$. This scheme has been optimized and tested in ref. 30 for Ra$^+$ and RaF excitation energies, and has also been successfully applied to calculated excitation energies in Ba$^+$ and BaF[56]. The contribution of the increased basis set described above is termed "+CBS" in the main text.

Correlation effects beyond the FS-RCCSD model have been calculated as the difference in transition energies calculated within the relativistic FS-RCCSDT and FS-RCCSD approaches using specially constructed compact natural contracted basis sets[36,43,44], correlating 27 RaF electrons and employing two-component (valence) GRECP Hamiltonian.

The atomic natural-like[71,72] compact basis[43] was constructed in such a way as to describe the $7S$, $6D$, $7P$, $8S$, $7D$, $5F$, and $8P$ states of the Ra$^+$ cation, which are relevant for the considered electronic states of RaF. For these states in Ra$^+$, scalar-relativistic 37e-CCSD(T) calculations were performed in an extended basis set, yielding correlated one-particle density matrices. For a better treatment of spin-orbit effects, we also calculated the one-particle density matrices corresponding to the occupied atomic spinors $\phi_p$ of the Ra$^+$ ion obtained in the two-component Hartree-Fock calculation. In the scalar atomic orbital (AO) basis function representation, these density matrices include not only diagonal "spin-up" and "spin-down" blocks, as in scalar-relativistic calculations, but also mixed-spin blocks. For example, one can consider the expression $\phi_p = \sum_\mu (C^\alpha_{p,\mu}\chi_\mu\alpha + C^\beta_{p,\mu}\chi_\mu\beta)$ in terms of scalar AO basis functions $\chi_\mu$ and spin functions $\alpha$, $\beta$, where $C^{\alpha/\beta}_{p,\mu}$ are expansion coefficients corresponding to two components of a given spinor. The contribution of such spinor to scalar AO density matrix will contain "mixed-spin" terms, such as $C^\alpha_{p,\mu}C^{\beta*}_{p,\nu}$. Density matrices in the AO

representation, obtained from both correlation calculations and two-component Hartree-Fock calculations, were averaged over electronic states, spin blocks, real and imaginary parts and projections of the AO basis functions. The resulting effective density matrix was then used as the matrix of the density operator in the AO representation. Corresponding eigenvectors with the highest eigenvalues (occupation numbers) were employed for constructing the compact basis set used in the RaF calculations. In total, the constructed compact basis set contained [$8s\,8p\,7d\,4f$] contracted basis functions for Ra, while for F we employed the aug-cc-pVDZ-DK basis set[73–75]. In principle, such a procedure could also be used within the Dirac-Coulomb Hamiltonian. However, at present one cannot use contracted Gaussian basis sets for heavy elements such as Ra in the employed implementation of the Dirac-Coulomb Hamiltonian. Another obstacle to the direct use of the Dirac-Coulomb Hamiltonian for the compact basis-set construction is the presence of serious practical limitations in the size of the original basis set that is used to construct correlated density matrices.

Lastly, the contributions of QED[30,42] as well as Gaunt inter-electron effects[41] were calculated at the FS-RCCSD level. To calculate the Gaunt contribution, we firstly solved self-consistent problems using 4-component Dirac-Coulomb-Gaunt and Dirac-Coulomb calculations, and then applied the X2C technique within the molecular mean-field approximation[41]. The final Gaunt contribution was obtained as the difference between the FS-RCCSD results with these two Hamiltonians. To calculate the QED contribution, we performed two FS-RCCSD calculations with the Dirac-Coulomb Hamiltonian. In the first, the model QED Hamiltonian was added after the self-consisted field but before the correlation stage of the calculation, while in the second, QED effects were not included. In these FS-RCC calculations, no virtual orbital cutoff was applied.

The excitation energies were calculated as the differences between total energies of the excited electronic states and the ground state at the internuclear distance 2.24 Å, which corresponds to the equilibrium internuclear distance of the electronic ground state of RaF following the scheme outlined above. However, to account for the difference in total energies at the equilibrium distance of each state, it was necessary to determine the equilibrium internuclear distances for all states (see Supplementary Table V). The obtained "non-verticality" contributions were added to the vertical "bulk" 69-electron FS-RCCSD excitation energies obtained in the large-scale calculations. To calculate these contributions, potential energy curves for all considered electronic states were calculated (see Suppl. Fig. 6).

Similarly to ref. 29, the potential energy curve for the electronic ground state was calculated within the two-component (valence) GRECP approach using the single-reference coupled cluster with single, double and perturbative triple cluster amplitudes CCSD(T) method. To obtain potential energy curves for excited electronic states as a function of the internuclear distance $R$, the excitation energy calculated at the FS-RCCSD level for a given value of $R$ was added to the energy of the electronic ground state for the same value of $R$. In these calculations, 37 electrons were included in the correlation treatment. For Ra, we used the [$25s\,17p\,12d\,9f\,4g\,2h$] basis set optimized for the GRECP calculations, while for F the uncontracted Dyall's AAEQZ basis set[37] was employed.

To characterize molecular terms in the $\Lambda S$ coupling scheme, the mean values of the electron orbital angular momentum projection operator on the molecular axis were calculated at the FS-RCCSD level (within the finite-field approach) and rounded up to an integer using the code to compute corresponding matrix elements developed in ref. 44.

In the uncertainty estimations, it is considered that the CBS correction has an uncertainty of 50%. The calculations did not take into account the gauge-dependent term of the Breit interaction (in the zero-frequency limit). According to the data for the Ra$^+$ ion[30,76], the contribution of the remaining part of the Breit interaction to transition energies in Ra$^+$ is within 20% of the Gaunt interaction effect.

Thus, the uncertainty due to the excluded part of the Breit interaction is estimated as 20% of the Gaunt-effect contribution. The accuracy of the model QED operator approach is high[42,77]. Therefore, the uncertainty of the QED correction is suggested to be in the order of 20%. The uncertainty of the high-order correlation effects (beyond the FS-RCCSDT model) can be estimated by comparing the FS-RCCSDT result with the single-reference coupled cluster with single, double, triple and perturbative quadruple cluster amplitudes, CCSDT(Q)[68,69]. According to our calculations performed for states of RaF with near single-reference character, the correlation corrections to FS-RCCSD calculated with the FS-RCCSDT and CCSDT(Q) methods agree within 60%. To be more conservative, in the uncertainty estimation we set the uncertainty of the higher-order correlation contribution to 75% of its value. The final uncertainty estimation of the theoretical electronic excitation energies was calculated as the square root of the sum of squares of the uncertainties described above for each state.

To obtain the composition of the RaF molecular electronic states in terms of Ra$^+$ states, the expectation values of the projectors on the corresponding Ra$^+$ states were calculated, e.g., $|7s_{1/2}\rangle\langle7s_{1/2}|$, $|7p_{1/2}\rangle\langle7p_{1/2}|$, etc., where the one-electron functions $|7s_{1/2}\rangle$ and $|7p_{1/2}\rangle$ were obtained as low-lying virtual orbitals in the atomic relativistic Hartree-Fock calculation. These composition calculations were performed within the compact basis set described above. To calculate the mean values of these projectors, we firstly computed the matrix elements of these operators over molecular spinors and then applied a finite-order expansion technique[32] to obtain expectation values of one-electron operators over the FS-RCCSD wave function. Due to the intrinsic semi-quantitative character of the procedure used, each contribution was rounded to tens of a percent.

### Reporting summary

Further information on research design is available in the Nature Portfolio Reporting Summary linked to this article.

### Data availability

The spectra generated and analyzed in this study have been deposited in the Zenodo repository at https://doi.org/10.5281/zenodo.8196151. The raw data are available upon request to the authors. Further information about data analysis is provided in the Supplementary Information. Source data are provided with this paper.

### Code availability

The analysis code used to pre-process raw data prior to contour fitting with PGOPHER can be provided upon request to the authors.

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

## Acknowledgements

We thank the ISOLDE collaboration for funding support and the ISOLDE technical teams for their work and assistance. This project has received funding from the European Union's Horizon Europe Research and Innovation programme under Grant Agreement No. 101057511. The research leading to these results has received funding from the European Union's Horizon 2020 research and innovation programme under Grant Agreement No. 654002. Financial support from FWO, as well as from the Excellence of Science (EOS) programme (No. 40007501) and the KU Leuven project C14/22/104, is acknowledged. The STFC consolidated grants ST/V001116/1 and ST/P004423/1 and the FNPMLS ERC grant agreement 648381 are acknowledged. SGW, RFGR, and SMU acknowledge funding by the Office of Nuclear Physics, U.S. Department of Energy Grants DE-SC0021176 and DE-SC002117. AAB, TFG, RB, and KG acknowledge funding from the Deutsche Forschungsgemeinschaft (DFG) - Projektnummer 328961117 - SFB 1319 ELCH. We thank the Center for Information Technology at the University of Groningen for their support and for providing access to the Peregrine high performance computing cluster. MAu, MN, AR, JWa, and JWe acknowledge funding from the EU's H2020-MSCA-ITN Grant No. 861198 -LISA'. DH acknowledges financial support from the Swedish Research Council (2020-03505). JL acknowledges financial support from STFC grant ST/V00428X/1. LFP acknowledges the support from the Dutch Research Council (NWO) project number VI.C.212.016 and the Slovak Research and Development Agency projects APVV-20-0098 and APVV-20-0127.

## Author contributions

M.A.-K., S.G.W., L.V.S., and G.N. led the manuscript preparation. M.A.-K., S.G.W., Á.K., A.A.B., O.A., M.Au, S.W.B., I.B., J.B., R.B., C.B., M.L.B., A.Br., K.C., T.E.C., R.P.d.G., A.D., C.M.F.Z., R.W.F., K.T.F., S.F., R.F.G.R., K.G., S.G., T.F.G., D.H., R.H., P.I., A.A.K., S.K., L.L., P.L., J.L., Y.C.L., K.M.L., A.McG., W.C.M., G.N., M.N., L.N., H.A.P., A.R., J.R.R., S.R., E.S., S.-M.U., B.v.d.B., Q.W., J.Wa., J.We., X.F.Y., and C.Z. performed the experiment. L.V.S. R.B., A.Bor., K.G., T.A.I., A.A.K., L.F.P., and C.Z. performed the calculations. M.A.-K., C.M.F.Z., and A.A.B. performed the data analysis. All coauthors participated in editing and revising the manuscript.

## Competing interests

The authors declare no competing interest.

## Additional information

[1]Experimental Physics Department, CERN, Geneva, Switzerland. [2]KU Leuven, Instituut voor Kern- en Stralingsfysica, Leuven, Belgium. [3]Blackett Laboratory, Centre for Cold Matter, London, UK. [4]Department of Physics, Massachusetts Institute of Technology, Cambridge, MA, USA. [5]Laboratory for Nuclear Science, Massachusetts Institute of Technology, Cambridge, MA, USA. [6]Institut für Optik und Atomare Physik, Technische Universität Berlin, Berlin, Germany. [7]Laboratory for Astrophysics, Institute of Physics, University of Kassel, Kassel, Germany. [8]Systems Department, Geneva, Switzerland. [9]Department of Chemistry, Johannes Gutenberg-Universität Mainz, Mainz, Germany. [10]School of Physics and State Key Laboratory of Nuclear Physics and Technology, Peking University, Beijing, China. [11]TRIUMF, Vancouver, BC, Canada. [12]Fachbereich Chemie, Philipps-Universität Marburg, Marburg, Germany. [13]Department of Physics and Astronomy, The University of Manchester, Manchester, UK. [14]Van Swinderen Institute of Particle Physics and Gravity, University of Groningen, Groningen, Netherlands. [15]Department of Chemistry, Massachusetts Institute of Technology, Cambridge, MA, USA. [16]Photon Science Institute, The University of Manchester, Manchester, UK. [17]Laboratoire Irène Joliot-Curie, Orsay, France. [18]University Paris-Saclay, Orsay, France. [19]Department of Physics, University of Gothenburg, Gothenburg, Sweden. [20]Department of Physics, University of Jyväskylä, Jyväskylä, Finland. [21]Department of Physical and Theoretical Chemistry, Faculty of Natural Sciences, Comenius University, Bratislava, Slovakia. [22]School of Nuclear Science and Technology, Lanzhou University, Lanzhou, China. [23]GSI Helmholtzzentrum für Schwerionenforschung GmbH, Darmstadt, Germany. [24]Department of Chemistry - TRIGA Site, Johannes Gutenberg-Universität Mainz, Mainz, Germany. [25]Affiliated with an institute covered by a cooperation agreement with CERN. ✉e-mail: m.athkak@cern.ch; wilkinss@mit.edu; gerda.neyens@kuleuven.be

