## [Transparent Peer Review file · Nature Communications]

Electron correlation and relativistic effects in the excited states of radium monofluoride

Corresponding Author: Dr Michail Athanasakis-Kaklamanakis

Version 0:

Reviewer comments:

Reviewer #1

(Remarks to the Author)

To perform experiments with short-lived radioactive atoms is by no ways trivial. The authors present a very detailed experimental and theoretical study on RaF. Such exotic radioactive molecules could reveal physics beyond the Standard Model. Electronic excited states were measured supported by state-of-the-art theoretical calculations, at the limit of what one can do with such multi-electron systems. The paper is well written and imminently suitable for Nature Communications. The justification for this is that the work becomes important for future measurements to probe the physics beyond the Standard Model. I recommend publication as it is except for one odd sentence. I disagree with the statement "This model is capable of explaining almost all experimental observations" as it is definitely not because of the missing gravitational interactions. Please correct or rephrase.

Reviewer #2

(Remarks to the Author)

The present manuscript is a joint experimental and theoretical endeavor to identify the lower states of the radioactive RaF molecule, which holds interest in spectroscopic tests of fundamental physics. We overall find this to be solid work, well worthy of publication of Nature Communications. However, some sharpening/corrections are necessary:

1) At several places in the manuscript we find the qualifier "fully relativistic". This has become standard terminology referring to 4-component relativistic calculations. However, the two-electron part of the 4-component electronic Hamiltonian is strictly not Lorentz invariant, so we would recommend using more precise language, also because the 4-component DC or DCG Hamiltonian was only used at the HF-level, whereas the correlated calculations were based on the X2C molecular mean-field approach, or relativistic effective pseudopotentials.

2) line 65: offers -> offer

3) If we strip the sentence starting on line 67 down to the essentials, it reads "calculations can be studied", which is meaningless. Please sharpen.

4) Line 189: "below the total theoretical uncertainty" refers to the SI, but it would be better if a number is also given directly in the text.

5) Starting on line 195 is a sentence about the CBS correction which refers to the SI, but the information is in fact found in the method section.

6) We are afraid that we do not find figure 3 particularly useful. No correlated calculation actually uses the Hamiltonians DC, DCG and DCG+QED. We believe it would be more useful to have a Table summarizing the various computational protocols, some of which provide correction to be added to previous numbers.

7) A general remark is that the main text spend a lot of time discussing the convergence of the various computational approaches towards the final nicely accurate results. Although it is nice to know that protocol is needed to achieve the end level of accuracy, we believe that most of this discussion fits better in the SI. We instead recommend in the main text to focus on the nature of the states of RaF.

8) We also note that Figures 5c-f are not even discussed in the text, so these should as a minimum be placed in the SI.

9) We also feel that there is some arbitrariness in what method information is given in the Method section and in the SI. We believe that the method section should provide all necessary steps towards the calculation of the the T0 excitation energies reported in Table I. For instance, it is quite important information to know that a CCSD(T) ground state curve was shifted by FSRCCSD energies.

- 10) Please use the same (italic) font for the energy E of eq (1) and in the subsequent line.
- 11) Line 301 starts quite abruptly discussing Rydberg series. We recommend a few lines motivating the subsequent discussion, notably that the ground and excited states of RaF may be modelled by a single electron outside closed shells. Another point is that with so few states it is not easy to identify the Rydberg series. We also note that the whole discussion is in a non-relativistic framework, which may require some justification.
- 12) Line 503 talks about the spectrum of a state, which sounds somewhat awkward since a spectrum is composed of transitions.
- 13) Line 570 refers to "Dirac-Hartree-Fock" which is possibly an unfortunate mix of Hamiltonian and method name; the authors do not write "Dirac-coupled-cluster".
- 14) On line 602 capital "L" should be replaced by small letter (one-electron quantity).
- 15) The section starting on line 633 about the construction of compact natural contracted bases needs sharpening. The construction is not very clear from just reading this paragraph. It sounds similar to the construction of ANO basis sets known from the MOLCAS package.
- 16) Starting at line 666 we learn that Gaunt and QED corrections were included at the FS-RCCSD level. However, it is not clear if these were included self-consistently at the SCF-level. Please sharpen.
- 17) The projection procedure (line 669++) used to obtain the composition of states in Table VI is not clear. We learn that projectors were built from HF atomic orbitals, but how were these applied to a CC wave function? Also, the caption of Table VI refers to "composition of states", whereas the Table reads more like the composition of the HOMOs.
- 18) In the SI (line 156++) we read that the calculations did not take into account the retardation part of the Breit interaction. We believe that this is rather what is often called the gauge-dependent term. Retardation would give a leading correction in c^{-3} .
- 19) Please check authors of ref. 3.
- 20) Ref. 8 has an erratum. Please add.
- 21) Please check titles of ref. 36 and 54. Proper capital letter are in bibtex assure by using double curly brackets for titles, e.g. title = {{...}}.
- 22) Please remove explicit web-link for ref. 66.

Reviewer #3

(Remarks to the Author)

In this manuscript, the authors report on a detailed spectroscopic study of the molecule RaF, using both experimental and theoretical methods. On the experimental side, they have observed a large number of excited electronic states at moderate resolution, allowing them to determine (in collaboration with theory) the angular momentum quantum numbers for a number of excited states. This work builds on the already extensive knowledge of alkaline-earth monofluorides by adding insight to the Ra-containing congener. By and large, the electronic structure appears to mirror that of very well-understood species such as CaF, SrF, and BaF. Generally speaking, the manuscript is well written, the experimental and theoretical methods appear well described and justifiable, and sufficient detail for replication is included. The authors have done an admirable job in their analysis and it is difficult to find fault with any of it. This is a nice manuscript and I have very few quibbles with it. That said, given the strong similarities to many decades of work studying alkaline-earth monohalides, perhaps publication in a more specialized community (J. Mol. Spec. or J. Chem. Phys.) would be more appropriate.

I do have a few minor questions for the authors that they might consider addressing in their manuscript:

- 1) Could the authors comment specifically on what limits their resolution (presumably the high temperature of the molecular beam) and what measures will be taken to improve that resolution? They claim that one benefit of these studies is to enable future high-resolution spectroscopy, but it is difficult to see how likely those experiments are in the current experimental setup without a valid method to reduce the molecular internal temperature.
- 2) The authors mention and cite some formative literature on ligand-field approaches to understanding alkaline-earth monohalides. I wonder whether their manuscript would be improved by actually applying a ligand-field approach to the presently studied RaF molecule. This could add considerable intuition to the complex ab initio methods.
- 3) The manuscript would be improved if in addition to theoretical term values (like those shown in Fig. 2) the authors reported some potential energy curves at a reasonable level of theory. It appears from the text that these curves have been calculated, so I wonder if the authors might consider adding a plot to either the main text or supplement.

Reviewer #4

(Remarks to the Author)

Version 1:

Reviewer comments:

Reviewer #2

(Remarks to the Author)

The revised manuscript reads better. However, the authors are aiming for a high-impact journal and so we feel as reviewers that we should ask for the highest standard not only in the actual work, but also in the presentation of it, and here there is still room for improvement.

1) In our previous review we suggested that figure 3 is not very useful and could be removed. Specifically, the authors write that they prefer to retain the figure "as it illustrates for a broad audience the 'degrees of freedom' considered in the calculation. Here our answer is that this is only partially true. An important is also the number of correlated electrons, which is not conveyed by the figure. Also, when we follow the sequence A -> E of calculations, we get the impression that they were all calculated in the same manner, simply by adjusting the three indicated parameters of calculations. However, this is not true. Beyond a certain point, the authors turn to composite calculations, e.g. including the effect of triples using a smaller basis etc. A reader may also want to go back to figure 2 (which is otherwise very nice) and see what data points correspond to protocols (A,B,C,D,E) and find that they are only partially represented. We suggest that the authors think hard about how to present their nice work in an optimal manner. The authors are free to keep Figure 3, but text should clarify that it does not directly describes the computational procedure that they follow.

2) Two remarks to the same sentence of the abstract:

- "excitation energy of excited states" sounds like a pleonasm to us

- "is studied, found to" is better replaced by something like "is studied and found to"

3) In the caption to figure 2 we read "The wavenumber scaling...". This sounds like a scaling with wave number, which is incorrect. Maybe something like "the wavenumber scale" ? Please sharpen.

4) The authors give statistical and systematic errors in round and square brackets, respectively. An inadvert reader may confuse the latter with a citation. However, more bothersome is that the authors on line 309 use another notation (that we like better). Please be consistent.

5) We already pointed out in our previous review that the authors should give more weight to a description of the calculated states, and they have partially listened to this. The HOMO of RaF is an essentially non-bonding Ra 7s orbital and the lower states are obtained by atomic-like excitations out of this. We can see this quite clearly from Table VI. We are not necessarily suggesting to move this Table into the main text, but a Table in the main text could accomodate information about the nature of the states, which may be helpful for understanding better aspects of the calculations as well as further modelling, e.g. the Rydberg series.

6) For instance, the authors note that the 27e-T correction is more important for the higher excited states. However, this could to a large extent be a relaxation effect, since the calculations employ the orbitals of the RaF cation.

7) On line 283 the authors note that, in contrast, the QED_effect is more important for lower excited states and write that "it decreases in importance as the average distance of the valence electrons from the radium nucleus increases for greater excitation energies". Is this an explanation or a hypothesis ? Did the authors check the validity of this statement ?

8) It would be nice if the values of the points of the potential curves in Figure VI of the Supplementary Material are also given. We believe that if someone wants to reproduce these computations, or to improve them, it would be very helpful to see the actual numbers.

Reviewer #3

(Remarks to the Author)

I would like to thank the authors for addressing my concerns. Despite my concerns that the work is largely a continuation of the light alkaline-earth congeners, the opinions of the editors and other referees convince me that there are no further roadblocks I see to acceptance of this manuscript.

Reviewer #4

(Remarks to the Author)

Version 2:

Reviewer comments:

Reviewer #2

(Remarks to the Author)

The second revision of the manuscript reads very well. We recommend that the manuscript be accepted, albeit with two minor modifications that do not require further review:

1) We believe that the inclusion of Table III in the main text is a good idea. However, now it's presence should be better exploited. We propose that Table II and III are swapped. Next, following the sentence: "As in the lighter alkaline-earth monohalides, all bound electronic states in RaF belong to Rydberg series of an unpaired electron centered on a RaF+ core, which converge to the ground state of RaF+." the authors could add something like "The composition of the orbital occupied by the unpaired electron is given in Table II". Getting information about the nature of the excited states at this point will greatly help the reader.

2) Line 153: "The search for the initial discovery" sounds somewhat strange. Please sharpen.

Reviewer #4

(Remarks to the Author)

Reviewer comment

Our response

Original text

Revised text

Reviewer #1 (Remarks to the Author):

To perform experiments with short-lived radioactive atoms is by no means trivial. The authors present a very detailed experimental and theoretical study on RaF. Such exotic radioactive molecules could reveal physics beyond the Standard Model. Electronic excited states were measured supported by state-of-the-art theoretical calculations, at the limit of what one can do with such multi-electron systems. The paper is well written and imminently suitable for Nature Communications. The justification for this is that the work becomes important for future measurements to probe the physics beyond the Standard Model. I recommend publication as it is except for one odd sentence.

We thank the reviewer for the positive comments and support for publication.

I disagree with the statement "This model is capable of explaining almost all experimental observations" as it is definitely not because of the missing gravitational interactions. Please correct or rephrase.

We are unable to locate where in the main manuscript or the supplementary information this statement appears. Is it possible for the reviewer to provide us with the line number? We will gladly proceed with the requested change afterwards. Alternatively, since this statement does not appear in the re-submitted manuscript, we hope that no further changes are necessary.

Reviewer #2 (Remarks to the Author):

The present manuscript is a joint experimental and theoretical endeavor to identify the lower states of the radioactive RaF molecule, which holds interest in spectroscopic tests of fundamental physics. We overall find this to be solid work, well worthy of publication of Nature Communications.

We thank the reviewer for their effort in the review of our manuscript and the positive comments in support of its publication.

However, some sharpening/corrections are necessary:

1) At several places in the manuscript we find the qualifier "fully relativistic". This has become standard terminology referring to 4-component relativistic calculations. However, the two-electron part of the 4-component electronic Hamiltonian is strictly not Lorentz invariant, so we would recommend using more precise language, also because the 4-component DC or DCG Hamiltonian was only used at the HF-level, whereas the correlated calculations were based on the X2C molecular mean-field approach, or relativistic effective pseudopotentials.

In line with the reviewer's comment, we have replaced "fully relativistic" by "relativistic" throughout the manuscript.

2) line 65: offers -> offer

The word in line 65 has been changed.

3) If we strip the sentence starting on line 67 down to the essentials, it reads "calculations can be studied", which is meaningless. Please sharpen.

We agree with the reviewer that the sentence in lines 67-70:

For these molecules, calculations using a fully relativistic Hamiltonian with high-order corrections and the inclusion of electron-correlation effects can be studied with up to a single, non-bonding valence electron.

is unnecessary, and it has been removed. The paragraph starting in line 64 is now merged with the next one, previously starting in line 71.

4) Line 189: "below the total theoretical uncertainty" refers to the SI, but it would be better if a number is also given directly in the text.

The text leading to line 189 is modified as follows:

Adding the remaining 28 electrons that correspond to the 1s–3d shells of Ra, thus including all 97 RaF electrons in the correlation space, modified the level energies by up to 2 cm^{-1} (see Table III in Supplementary Information), which is significantly below the total theoretical uncertainty, reported in Table I for each state.

5) Starting on line 195 is a sentence about the CBS correction which refers to the SI, but the information is in fact found in the method section.

The reference to the additional details is corrected from *Supplementary Information* to *Methods*.

6) We are afraid that we do not find figure 3 particularly useful. No correlated calculation actually uses the Hamiltonians DC, DCG and DCG+QED. We believe it would be more useful to have a Table summarizing the various computational protocols, some of which provide correction to be added to previous numbers.

In the Supplementary Information, Table 2 shows how the final theoretical values are obtained by combining different contributions. We have added a reference to this table in the main text (line 219+), to facilitate access to the information for the readers, which reads:

The contribution of different corrections to the final theoretical transition energies are provided in Table II of the Supplementary Information.

We prefer to retain Figure 3, “Schematic illustration of the electronic calculation scheme...” as it illustrates for a broad audience the “degrees of freedom” considered in the calculations. We understand that this is not particularly interesting to a specialist. Considering the broad audience that may be interested in our work, including experimentalists in AMO, nuclear physics, and nuclear chemistry, we consider such a figure helpful. In line with the reviewer’s comment on the use of the Hamiltonian, we have replaced the terms “DC” and “DCG” with “X2C-DC” and “X2C-DCG” to avoid confusion.

We also note that in our tests, the transition energies of RaF obtained in FS-RCCSD calculations with the extAE3Z basis set, performed for the 4-component Dirac-Coulomb and also X2C-transformed (X2C-DC) Hamiltonians, agree within 0.2 cm^{-1} for all considered electronic states.

7) A general remark is that the main text spend a lot of time discussing the convergence of the various computational approaches towards the final nicely accurate results. Although it is nice to know that protocol is needed to achieve the end level of accuracy, we believe that most of this discussion fits better in the SI. We instead recommend in the main text to focus on the nature of the states of RaF.

We agree with the reviewer that the text should be focused on the states of RaF and our understanding of the electronic structure as much as possible. We also agree that the paragraphs that describe the different steps taken to improve the Hamiltonian, correlation treatment, and basis-set quality temporarily take the focus away from the structure of RaF, and place it on the calculation details.

The text was placed at that point of the main text after considering the diverse audience that may be interested in this work, which includes experimentalists in AMO, nuclear physics – particularly scientists working at facilities that produce short-lived atoms like radium, and who are interested in radioactive molecular developments – and nuclear chemistry. As such, we felt that a few short paragraphs that explain the different theoretical corrections are necessary early in the *Results and Discussion* section.

While these paragraphs could be placed in the *Methods*, we considered that, for many readers, the absence of a short explanation of the calculations early in the main text would make it difficult to fully appreciate the message we want to convey in the *Results and Discussion*. This was confirmed to us by colleagues who did not participate in this work, but who kindly provided comments on our work to assist us in improving the readability of the text.

As a result, if the reviewer agrees, we would like to keep these paragraphs as they are. While they momentarily take away from the focus of the article, we believe that they are helpful for parts of the intended audience to understand the results and their discussion more easily.

8) We also note that Figures 5c-f are not even discussed in the text, so these should as a minimum be placed in the SI.

Figure 5 of the main text now only includes subfigures (a) and (b), while (c)-(f) are moved to the Supplementary Information as Figure 5.

9) We also feel that there is some arbitrariness in what method information is given in the Method section and in the SI. We believe that the method section should provide all necessary steps towards the calculation of the the T0 excitation energies reported in Table I. For instance, it is quite important information to know that a CCSD(T) ground state curve was shifted by FSRCCSD energies.

We have now concentrated all the information on the calculations that was split between *Methods* and *Supplementary Information* and added it to the corresponding subsection in *Methods*. While this lengthens somewhat the *Methods*, it was done also to abide by the journal's specifications that the *Methods* should contain all information necessary to understand and replicate the results.

10) Please use the same (italic) font for the energy E of eq (1) and in the subsequent line.

The font for E has been corrected in Eq. (1) and made into italic.

11) Line 301 starts quite abruptly discussing Rydberg series. We recommend a few lines motivating the subsequent discussion, notably that the ground and excited states of RaF may be modelled by a single electron outside closed shells. Another point is that with so few states it is not easy to identify the Rydberg series. We also note that the whole discussion is in a non-relativistic framework, which may require some justification.

We have added the following paragraph before Line 301:

As in the lighter alkaline-earth monohalides, all bound electronic states in RaF belong to Rydberg series of an unpaired electron centered on a RaF⁺ core, which converge to the ground state of RaF⁺. This simple picture provides a way to understand the electronic structure of the molecule even at high excitation energy, with non-relativistic multichannel quantum defect theory based on this picture being very successful for the lighter homologs [<https://doi.org/10.1063/1.473124> ; <https://doi.org/10.1063/1.3565967>].

12) Line 503 talks about the spectrum of a state, which sounds somewhat awkward since a spectrum is composed of transitions.

The sentence is changed as:

The spectrum of the previously reported transition from the ground state to the A ²Π_{1/2} state[...].

13) Line 570 refers to "Dirac-Hartree-Fock" which is possibly an unfortunate mix of Hamiltonian and method name; the authors do not write "Dirac-coupled-cluster".

"Dirac-Hartree-Fock" – one also uses even "Dirac-Fock" – is an established term that has been widely used in scientific literature for decades. We can provide the following articles as examples:

<https://journals.aps.org/pr/abstract/10.1103/PhysRevA.9.1486>;

<https://www.sciencedirect.com/science/article/abs/pii/S0375947401011393> ;

We would therefore prefer to keep the term “Dirac-Hartree-Fock” in the *Methods* section.

14) On line 602 capital "L" should be replaced by small letter (one-electron quantity).

The capital *L* has been replaced by lower-case *l* across the whole paragraph.

15) The section starting on line 633 about the construction of compact natural contracted bases needs sharpening. The construction is not very clear from just reading this paragraph. It sounds similar to the construction of ANO basis sets known from the MOLCAS package.

Indeed, the employed approach is similar to the ANO case. We have attempted to have this reflected in the opening phrase of the paragraph, “The atomic natural-like...”. We have now added two references [P.-O. Widmark, P.-A. Malmqvist, and B. O. Roos, *Theor. Chim. Acta* 77, 291–306 (1990); J. Almlöf and P. R. Taylor, in *Advances in Quantum Chemistry Volume 22* (Elsevier, 1991) pp. 301–373] to this sentence (Line 650).

We note that we use our own version (and code) of this approach, which differs somewhat in practice; therefore, the method used is described in this paragraph. We have added further details to the text, in lines 663-669 and 679-682.

16) Starting at line 666 we learn that Gaunt and QED corrections were included at the FS-RCCSD level. However, it is not clear if these were included self-consistently at the SCF-level. Please sharpen.

We included Gaunt contributions self-consistently already at the SCF-stage. The model QED operator was added after SCF but before the FS-RCCSD stage due to technical reasons, however to allow relaxation effects which can be accounted for by the exp(T1) operator of the RCC method, we did not impose any cutoff for virtual orbitals. We have added the following text:

“To calculate the Gaunt contribution, we firstly solved self-consistent problems using 4-component Dirac-Coulomb-Gaunt and Dirac-Coulomb calculations and then applied the X2C technique within the molecular mean-field approximation. The final Gaunt contribution was obtained as the difference between FS-RCCSD results with these two Hamiltonians. To calculate the QED contribution, we performed two FS-RCCSD calculations with the Dirac-Coulomb Hamiltonian. In the first, the model QED Hamiltonian was added after the self-consistent field but before the correlation stage of the calculation, while in the second, QED effects were not included. In these FS-RCC calculations, no virtual orbital cutoff was applied.”

17) The projection procedure (line 669++) used to obtain the composition of states in Table VI is not clear. We learn that projectors were built from HF atomic orbitals, but how were these applied to a CC wave function ? Also, the caption of Table VI refers to "composition of states", whereas the Table reads more like the composition of the HOMOs.

The projector is an one-electron operator. Thus, we calculated its expectation value over the FS-RCCSD wavefunction. This corresponds to a calculation of occupation of a corresponding valence atomic orbital (spinor) using correlated molecular density matrix. We have added the following text (Line 783+):

“To calculate the mean values of these projectors, we firstly computed the matrix elements of these operators over molecular spinors and then applied a finite-order expansion technique [A. Zaitsevskii, et al, Molecular Physics, e2236246 (2023)] to obtain expectation values of one-electron operators over the FS-RCCSD wave function.”

Additionally, we have modified the caption of Table VI in the Supplementary Information as follows:

“Composition of states in RaF in terms of Ra⁺ valence electron configurations. The composition is calculated as the mean value of the projectors onto the one-electron atomic orbitals of the Ra⁺ cation over the FS-RCCSD wave function of RaF (see main text). Only contributions with a relative impact of $\geq 10\%$ are shown.”

18) In the SI (line 156++) we read that the calculations did not take into account the retardation part of the Breit interaction. We believe that this is rather what is often called the gauge-dependent term. Retardation would give a leading correction in c^{-3} .

In line with the reviewer’s comment, we have replaced “retardation” by “gauge-dependent”.

19) Please check authors of ref. 3.

The reference has been updated to have the correct author list.

20) Ref. 8 has an erratum. Please add.

A reference to the erratum has been added.

21) Please check titles of ref. 36 and 54. Proper capital letter are in bibtex assure by using double curly brackets for titles, e.g. title = {{...}}.

The capitalization has been fixed.

22) Please remove explicit web-link for ref. 66.

The web link has been removed.

Reviewer #3 (Remarks to the Author):

In this manuscript, the authors report on a detailed spectroscopic study of the molecule RaF, using both experimental and theoretical methods. On the experimental side, they have observed a large number of excited electronic states at moderate resolution, allowing them to determine (in collaboration with theory) the angular momentum quantum numbers for a number of excited states. This work builds on the already extensive knowledge of alkaline-earth monofluorides by adding insight to the Ra-containing congener. By and large, the electronic structure appears to mirror that of very well-understood species such as CaF, SrF, and BaF. Generally speaking, the manuscript is well written, the experimental and theoretical methods appear well described and justifiable, and sufficient detail for replication is included. The authors have done an admirable job in their analysis and it is difficult to find fault with any of it. This is a nice manuscript and I have very few quibbles with it.

We would like to thank the reviewer for their effort and positive comments on our manuscript.

That said, given the strong similarities to many decades of work studying alkaline-earth monohalides, perhaps publication in a more specialized community (J. Mol. Spec. or J. Chem. Phys.) would be more appropriate.

We consider that our manuscript will be of interest to a wide range of researchers in AMO theory and experiment, nuclear physics, and nuclear chemistry. As a result, we view *Nature Communications* as a suitable journal due to its broad audience.

I do have a few minor questions for the authors that they might consider addressing in their manuscript:

1) Could the authors comment specifically on what limits their resolution (presumably the high temperature of the molecular beam) and what measures will be taken to improve that resolution? They claim that one benefit of these studies is to enable future high-resolution spectroscopy, but it is difficult to see how likely those experiments are in the current experimental setup without a valid method to reduce the molecular internal temperature.

The present measurements were obtained using broadband lasers with a linewidth of a few GHz. Using narrowband pulsed lasers (linewidth less than 50 MHz) and with the current experimental setup, a spectroscopic resolution of ~100 MHz has been achieved in ²²⁶RaF (see [nature.com/articles/s41567-023-02296-w](https://www.nature.com/articles/s41567-023-02296-w)), and in a recent experiment we successfully improved the resolution by another factor of 2 (the results are unpublished). This is possible thanks to the compression of the Doppler broadening that is inherent in collinear spectroscopy using fast beams.

2) The authors mention and cite some formative literature on ligand-field approaches to understanding alkaline-earth monohalides. I wonder whether their manuscript would be improved by actually applying a ligand-field approach to the presently studied RaF molecule. This could add considerable intuition to the complex ab initio methods.

We have indeed considered applying a ligand-field model to analyze the observed excited states. We decided to reserve this for a future work, as we would like to observe electronic states at even higher excitation energies, adding more states to each Rydberg series, before applying such a

model to the electronic structure of RaF. This would significantly improve the success of the ligand-field model. We thank the reviewer for the suggestion.

3) The manuscript would be improved if in addition to theoretical term values (like those shown in Fig. 2) the authors reported some potential energy curves at a reasonable level of theory. It appears from the text that these curves have been calculated, so I wonder if the authors might consider adding a plot to either the main text or supplement.

We have now added potential energy curves for all studied electronic states as Figure 6 in the Supplementary Information.

Reviewer #4 (Remarks to the Author):

We would like to thank the reviewer for their effort.

Dear Editor,

We are very thankful for your continued work on the review of our manuscript *Pinning down electron correlations in RaF via spectroscopy of excited states and high-accuracy relativistic quantum chemistry*. Please find below a point-by-point response to the reviewers' comments.

Kind regards,

Michail Athanasakis-Kaklamanakis on behalf of the authors of NCOMMS-24-54400

Reviewer comment

Our response

Original text

Revised text

Reviewer #2 (Remarks to the Author):

The revised manuscript reads better. However, the authors are aiming for a high-impact journal and so we feel as reviewers that we should ask for the highest standard not only in the actual work, but also in the presentation of it, and here there is still room for improvement.

We thank the reviewers for their effort, and we are especially grateful for their attention to detail, which has undoubtedly helped us improve our manuscript.

1) In our previous review we suggested that figure 3 is not very useful and could be removed. Specifically, the authors write that they prefer to retain the figure "as it illustrates for a broad audience the 'degrees of freedom' considered in the calculation. Here our answer is that this is only partially true. An important is also the number of correlated electrons, which is not conveyed by the figure. Also, when we follow the sequence A -> E of calculations, we get the impression that they were all calculated in the same manner, simply by adjusting the three indicated parameters of calculations. However, this is not true. Beyond a certain point, the authors turn to composite calculations, e.g. including the effect of triples using a smaller basis etc. A reader may also want to go back to figure 2 (which is otherwise very nice) and see what data points correspond to protocols (A,B,C,D,E) and find that they are only partially represented. We suggest that the authors think hard about how to present their nice work in an optimal manner. The authors are free to keep Figure 3, but text should clarify that it does not directly describes the computational procedure that they follow.

Following the reviewer's suggestion, we have removed Figure 3 and its in-text references.

2) Two remarks to the same sentence of the abstract:

- "excitation energy of excited states" sounds like a pleonasm to us

- "is studied, found to" is better replaced by something like "is studied and found to"

We have rephrased the two parts in the abstract as:

The role of high-order electron correlation and quantum electrodynamics effects in the excitation ~~energies~~ ~~energy of excited states~~ is studied; and found to be important for all states [...]

3) In the caption to figure 2 we read "The wavenumber scaling...". This sounds like a scaling with wave number, which is incorrect. Maybe something like "the wavenumber scale" ? Please sharpen.

We have rephrased the sentence in the caption as per the reviewer's suggestion. "The wavenumber scaling" is now written as "The wavenumber scale".

4) The authors give statistical and systematic errors in round and square brackets, respectively. An inadvertent reader may confuse the latter with a citation. However, more bothersome is that the authors on line 309 use another notation (that we like better). Please be consistent.

Following the reviewer's suggestion, we have changed the error notation into $(\text{stat})_{\text{sys}}$ in Table I and throughout the manuscript and the supplementary information.

5) We already pointed out in our previous review that the authors should give more weight to a description of the calculated states, and they have partially listened to this. The HOMO of RaF is an essentially non-bonding Ra 7s orbital and the lower states are obtained by atomic-like excitations out of this. We can see this quite clearly from Table VI. We are not necessarily suggesting to move this Table into the main text, but a Table in the main text could accommodate information about the nature of the states, which may be helpful for understanding better aspects of the calculations as well as further modelling, e.g. the Rydberg series.

Following the removal of Fig. 3 from the main text and thus the additional space, we have followed the reviewer's suggestion and moved the table with the RaF state composition in terms of Ra⁺ valence configurations to the main text. It now appears as Table III of the main text.

6) For instance, the authors note that the 27e-T correction is more important for the higher excited states. However, this could to a large extent be a relaxation effect, since the calculations employ the orbitals of the RaF cation.

It is primarily a question of terminology. In the conventional CC framework, single excitations in the CC wave operator, $\exp(T1)$, are typically associated with "relaxation" effects, as the $\exp(T1)$ operator connects two determinants and introduces substantial orbital insensitivity to the CC method [see, e.g., R.J. Bartlett, Rev. Mod. Phys. 79, 291 (2007)]. Double excitations are usually referred to as "correlation" (the situation is a bit more complicated in the FS-CC method in sector 0h1p). We believe that it is quite reasonable to categorize the contribution of triple excitations as correlation, even within the FS-CC method. On the other hand, since all cluster amplitudes are optimized within a single set of equations, there is certainly interference between relaxation and correlation. Therefore, we now also mention relaxation effects in the text (line 259):

This correction is larger in high-lying states (Fig. 3b) than in the low-lying ones (Fig. 3a), demonstrating the need for spectroscopic studies of electronic states far above the ground state to understand the role of correlations (and relaxation) in the electronic structure.

7) On line 283 the authors note that, in contrast, the QED_effect is more important for lower excited states and write that "it decreases in importance as the average distance of the valence

electrons from the radium nucleus increases for greater excitation energies". Is this an explanation or a hypothesis ? Did the authors check the validity of this statement ?

It is a hypothesis indeed. To avoid ambiguity, we have removed this statement.

"Figure 3c confirms that the relative impact of QED effects is indeed significant at low energies, but it decreases in importance as ~~the average distance of the valence electrons from the radium nucleus increases~~ for greater excitation energies."

8) It would be nice if the values of the points of the potential curves in Figure VI of the Supplementary Material are also given. We believe that if someone wants to reproduce these computations, or to improve them, it would be very helpful to see the actual numbers.

We agree that it is helpful if these numbers are explicitly given, as also indicated in the journal guidelines. We have added all these values in Table VII in the Supplementary Information.

Reviewer #3 (Remarks to the Author):

I would like to thank the authors for addressing my concerns. Despite my concerns that the work is largely a continuation of the light alkaline-earth congeners, the opinions of the editors and other referees convince me that there are no further roadblocks I see to acceptance of this manuscript.

We are grateful for the reviewer's effort and positive outlook on our work.

Reviewer #4 (Remarks to the Author):

We would like to thank the reviewer for their effort.

Reviewer comment

Our response

Original text

Revised text

Reviewer #2 (Remarks to the Author):

The second revision of the manuscript reads very well. We recommend that the manuscript be accepted, albeit with two minor modifications that do not require further review:

We thank the reviewer for their continued work, and the recommendation for acceptance.

1) We believe that the inclusion of Table III in the main text is a good idea. However, now it's presence should be better exploited. We propose that Table II and III are swapped. Next, following the sentence: "As in the lighter alkaline-earth monohalides, all bound electronic states in RaF belong to Rydberg series of an unpaired electron centered on a RaF⁺ core, which converge to the ground state of RaF⁺." the authors could add something like "The composition of the orbital occupied by the unpaired electron is given in Table II". Getting information about the nature of the excited states at this point will greatly help the reader.

We have implemented the reviewer's suggestion to swap the order in tables II and III, and we added the sentence they suggested in lines 297-299.

2) Line 153: "The search for the initial discovery" sounds somewhat strange. Please sharpen.

We have rephrased line 153 from: "The search for the initial discovery" to "The initial discovery" as per the reviewer's suggestion.